# Comprehensive Analysis of Ferroptosis Regulators with Regard to PD-L1 and Immune Infiltration in Low-Grade Glioma

**DOI:** 10.3390/ijms241612880

**Published:** 2023-08-17

**Authors:** Yuxuan Luan, Yuling Chen, Yilin Wang, Minfeng Shu

**Affiliations:** 1Department of Pharmacology, School of Basic Medical Sciences, Shanghai Medical College, Fudan University, 131 Dong’an Road, Shanghai 200032, China; 2Ministry of Education (MOE) & Ministry of Health (MOH) Key Laboratory of Medical Molecular Virology, School of Basic Medical Sciences, Shanghai Medical College, Fudan University, 131 Dong’an Road, Shanghai 200032, China; 3Department of Hepatic Surgery, Fudan University Shanghai Cancer Center, 207 Dong’an Road, Shanghai 200032, China; 4Department of Oncology, Shanghai Medical College, Fudan University, 131 Dong’an Road, Shanghai 200032, China

**Keywords:** ferroptosis, low-grade glioma, CD274, prognostic markers, noncoding RNA

## Abstract

The prognosis of low-grade glioma (LGG) is highly variable and requires more accurate predictors. Ferroptosis, a newly discovered programmed cell death, has been demonstrated to play a crucial role in some types of tumors. However, prognostic prediction based on ferroptosis-related genes (FRGs) and the influence on the tumor microenvironment (TME) in LGG remains elusive. We derived expression profiles for LGG from public databases. Based on the expression of 25 FRGs in LGG, two independent subtypes and a risk model were successfully constructed. Different methods were applied to assess the tumor heterogeneity, tumor microenvironment, and the prognostic value. In addition, a competing endogenous RNA (ceRNA) regulatory axis was constructed. The subtypes had independent tumor heterogeneity, tumor microenvironments, and prognoses. LPCAT3, SLC1A5, HSPA5, and NFE2L2 were identified as the potential prognostic FRGs. Based on these four FRGs, our risk model possesses excellent potential to predict prognosis and varied immune infiltration abundance. The ceRNA regulatory axis provides a potential therapeutic target for LGG. Our molecular subtypes, risk model, and ceRNA regulatory axis have strong immune prediction and prognostic prediction capabilities which could guide LGG treatment.

## 1. Introduction

Gliomas are the most common primary malignant tumor of the central nervous system (CNS) and are divided into four grades by the WHO classification scheme [1,2]. Low-grade glioma (LGG) comprises WHO grade II and III gliomas, and the median patient survival duration of LGG is about 7 years [3]. At present, the general treatment regimen for LGG is aggressive surgical resection, postoperative chemotherapy and/or radiotherapy. However, more than half of LGG patients develop high recurrence and drug-resistance over time, with immunosuppressive effects and poor prognosis [4]. Emerging evidence shows that, compared with traditional histology, the molecular genetics correlate more significantly with LGG prognosis. IDH1/2 mutations, 1p/19q codeletion, histone H3-K27M mutations, and C11orf95-RELA fusions are widely utilized as diagnostic biomarkers [5,6]. However, they cannot predict the prognosis of LGG precisely. Therefore, novel prognostic biomarkers for risk stratification should be investigated in detail.

Ferroptosis, a newly discovered form of programmed cell death (PCD), is characterized by massive lipid peroxidation accumulation and membrane damage [7]. The blockade of the cystine/glutamate antiporter system (system Xc-) and inhibition of glutathione peroxidase 4 (GPX4) can lead to an imbalance of redox homeostasis and can increase accumulation of iron-dependent lipid-ROS. Ultimately, excessive intracellular iron results in ferroptosis. Previous studies showed that the activation of ferroptosis could enhance the therapeutic efficacy of pancreatic, liver, and kidney cancer cells. The combination of ferroptosis activators (such as erastin) and anti-PD-L1 antibodies was reported to have synergistic effects and could suppress tumor growth both in vitro and vivo [8,9]. However, ferroptotic damage could also lead to a microenvironment delivering nutrients for cancer and can trigger inflammation-associated immunosuppression, thus accelerating cancer growth [10]. To date, there is still a lack of integrated understanding of ferroptosis in LGG, including the crosstalk between ferroptosis-related genes (FRGs) and the tumor immunity milieu.

Here, we comprehensively analyzed the expression profiles, prognosis, correlations with CD274, and the roles in the tumor microenvironment (TME) of FRGs in LGG. Based on the expression levels of FRGs, we clustered two subtypes with apparent tumor heterogeneity, and different TME, and CD274 expression levels, which improves the risk stratification and precision therapy for LGG patients. Subsequently, four genes (LPCAT3, SLC1A5, HSPA5, and NFE2L2) were identified as potential prognostic and immune infiltration-related FRGs. Their mRNA expression levels and correlations with CD274 were validated with glioma cell lines using quantitative real-time PCR (qRT-PCR). In addition, we conducted the least absolute shrinkage and selection operator (LASSO) regression and established a prognostic risk model to separate LGG patients with distinct survival outcomes. At last, we analyzed the competing endogenous RNA network of four prognostic FRGs. Our findings provide novel insights into the regulatory mechanisms associated with TME and the approaches for LGG immunotherapy.

## 2. Results

The workflow of the entire study is shown in Figure 1.

### 2.1. Expression Divergence of FRGs between LGG and Normal Tissues

According to the pivotal roles of ferroptosis, we defined 25 ferroptosis regulators derived from a previous study [11] as FRGs. To evaluate their biological functions, the expression patterns of the selected FRGs in LGG and normal tissues were systematically explored using the GEPIA database. Consequently, distinct expression levels of FRGs were observed in LGG and normal tissues (Figure 2A,B), including eight upregulated FRGs (CDKN1A, FANCD2, HSPA5, LPCAT3, NCOA4, NFE2L2, RPL8, and SLC1A5; *p* < 0.05) and two down-regulated FRGs (GLS2 and MT1G; *p* < 0.05). However, the expression levels of 15 genes (ACSL4, ALOX15, CARS1, CISD1, CS, EMC2, FDFT1, GPX4, HSPB1, SAT1, ATP5G3, DPP4, SLC7A11, TFRC, and ATL1; *p* > 0.05) showed no significant differences (Figure 2C). In addition, we investigated the correlations between the expression of FRGs, which revealed strong associations (Figure 2D). These findings suggested that FRGs might play a crucial role in regulating tumorigenesis and LGG development.

### 2.2. Consensus Clustering Analysis of FRGs Revealed Significant Differences in Baseline Characteristics and Survival between Two Patient Clusters

The Consensus Cluster Plus package of R software (v1.54.0) was used for consistency analysis. According to the expression levels of the selected FRGs and the proportion of ambiguous clustering measure, k = 2 was identified as the optimal clustering stability from k = 2 to 6 (Figure 3A,B; Appendix A). Therefore, we divided 513 LGG patients into two subtypes, which were named cluster 1 (C1, *n* = 422) and cluster 2 (C2, *n* = 91). To further confirm the clustering results, we applied the principal component analysis (PCA) method to analyze the gene expression profiles between the two subtypes, which revealed large differences (Appendix A). In contrast to cluster 1, cluster 2 had fifteen high-expressed FRGs (CDKN1A, HSPA5, SLC7A11, NFE2L2, MT1G, HSPB1, GPX4, FANCD2, SLC1A5, SAT1, TFRC, LPCAT3, DPP4, CARS1, and ATP5MC3), and eight low-expressed FRGs (CISD1, FDFT1, RPL8, NCOA4, GLS2, CS, ACSL4, and ATL1), while there were no differences in the expression of two FRGs (ALOX15 and EMC2) between the two clusters (Figure 3C). As shown in Table 1, patients in the distinct subtypes had significant differences in clinicopathological characters and prognosis. Patients in cluster 1 were younger, had a lower cancer grade, and received less radiation therapy (*p* < 0.0001), whereas no statistical differences in gender and race were found between the two clusters (*p* > 0.05). In addition, patients in cluster 1 had higher rates of overall survival (*p* < 0.001), progression-free survival (*p* < 0.001), disease-specific survival (*p* < 0.001), and disease-free survival (*p* < 0.05) than cluster 2 (Figure 3D). These results indicate that there is significant heterogeneity between the two subtypes of LGG patients.

### 2.3. Association of FRGs with CD274 Expression Level and Immune Cell Infiltration in LGG

To investigate the relationship between CD274 and ferroptosis in LGG, we evaluated the divergent expression in two subtypes and different histological grades. The expression level of CD274 was higher in cluster 2 patients (*p* < 0.0001; Figure 4A). In contrast to normal tissues, the expression level of CD274 increased with pathological grade (*p* < 0.0001; Figure 4B). We subsequently analyzed the association between CD274 and FRGs in LGG, and found that twelve FRGs (ALOX15, DPP4, EMC2, HSPA5, HSPB1, LPCAT3, MT1G, NFE2L2, SAT1, SLC1A5, SLC7A11, and TFRC) had a positive association with CD274, while six FRGs (CISD1, CS, FANCD2, FDFT1, NCOA4, and RPL8) had a negative association (Figure 4C). To further investigate the impact of FRGs on the TME of LGG, we explored the differences in immune cell infiltration of the two classified subtypes (Figure 4D). Cluster 1 possessed higher infiltration levels of naive B cells, plasma B cells, naive T cell CD4+, monocytes, activated mast cells, and resting mast cells, whereas cluster 2 was more associated with memory B cells, CD8+ T cells, memory resting CD4+ T cells, memory-activated CD4+ T cells, Tregs, Macrophage M1, Macrophage M2, and Neutrophils. These findings suggest that the two subtypes had apparent TME.

### 2.4. Four Key FRGs (LPCAT3, SLC1A5, HSPA5, and NFE2L2) Were Upregulated in Glioma Tissues

To further determine the key FRGs, we used an intersection of highly expressed FRGs in LGG that are linked to poor prognosis and positively correlated with the expression of CD274. As shown in Figure 5A, four genes (LPCAT3, SLC1A5, HSPA5, and NFE2L2) were identified as the potential prognostic FRGs. Given that LGG and GBM are both types of glioma, we analyzed the glioma expression data from the GEO database to further illustrate the expression of the four key FRGs. The results show that, compared with normal tissues, the four key FRGs were all significantly high-expressed in gliomas in the GSE16011 cohort (Figure 5B). Subsequently, we detected the mRNA expression levels of the four key FRGs in glioma cells and normal astrocytes cells. The qRT-PCR revealed that these regulators were all high-expressed in T98G and DBTRG-05MG (Figure 5C,D), and the depletion of these FRGs could reduce CD274 mRNA expression levels in glioma cells (Figure 5E,F). These results demonstrate that the four key FRGs (LPCAT3, SLC1A5, HSPA5, and NFE2L2) were highly expressed in LGG and were positively correlated with CD274 expression.

### 2.5. Correlation between Four Key FRGs and the TME Heterogeneity

To further evaluate the expression profiles of the four key FRGs in TME-related cells, we used two datasets (Glioma_GSE131928_10X and Glioma_GSE89567) of the TISCH2 database. The Glioma_GSE131928_10X dataset was reported to contain IDH-wildtype glioblastomas, while Glioma_GSE89567 dataset contains IDH-mutant gliomas [12,13]. In the Glioma_GSE131928_10X dataset, eight cell types were clustered. Mesenchymal-like malignant cells (MES-like malignant cells) were the most abundant cell type (n = 5203; Figure 6A,B). The expression level of HSPA5 was the highest among the MES-like malignant cells, while LPCAT3 and NFE2L2 were highest among the astrocyte-like malignant cells (AC-like malignant cells). However, SLC1A5 was mainly expressed in exhausted CD8+ T cells (Figure 6C,D). In the Glioma_GSE89567 dataset, only four cell types were found, including oligodendrocytes, AC-like malignant cells, oligodendrocyte-like malignant cells (OC-like malignant cells), and monocyte/macrophage cells (Mono/Macro cells). Among these cells, OC-like malignant cells exhibited the most abundant cell counts (n = 4645; Figure 6E,F). The expression levels of HSPA5, NFE2L2, and SLC1A5 were all highest in Mono/Macro cells, while LPCAT3 was highest in AC-like malignant cells (Figure 6G,H). These results suggest that the cell components and distributions of IDH-wildtype glioblastomas and IDH-mutant gliomas are quite distinct. The expression levels of the four key FRGs were also different in distinct cell types, which might lead to the heterogeneity of the LGG microenvironment.

### 2.6. Construction of the Prognostic Model of Four Key FRGs in LGG

LASSO regression analysis was performed to construct a prognostic gene model based on the four key FRGs (Figure 7A,B). The risk score = (0.0304) × HSPA5 + (0.2461) × SLC1A5 + (0.9957) × NFE2L2. Based on the risk score, LGG patients were separated into two subgroups. The risk score distribution, survival status, and the expression of four key FRGs are presented in Figure 7C. As the expression levels of the four key FRGs increased, the patients’ risk increased and the survival time decreased (Figure 7C). The Kaplan–Meier curve revealed that the LGG patients with high-risk scores had a worse overall survival probability than those with low-risk scores (median time = 5.2 years vs. 9.5 years, *p* < 0.0001; Figure 7D), with areas under curves (AUCs) of 0.744, 0.706, and 0.653 in the 1-year, 3-year, and 5-year ROC curves, respectively (Figure 7E).

### 2.7. Correlation between Four Prognostic FRGs’ Expression Levels and Immune Infiltration Levels in LGG

It is uncertain whether these FRGs can influence immune cell recruitment in the TME and affect the prognosis of LGG. Thus, we performed an analysis to examine the relationships between the four prognostic FRGs and the immune infiltration in LGG. The expression levels of these FRGs were positively associated with the infiltration of macrophages and natural killer (NK) cells, whereas they were negatively correlated with B cells, endothelial cells, CD4+ T cells, and CD8+ T cells (Figure 8A). A further risk score analysis showed that high macrophage infiltration, as well as low B cells, CD4+ T cells, and CD8+ T cells infiltration was positively associated with a high-risk score (Figure 8B). These results suggest that the four prognostic FRGs can influence the tumor immune infiltration, and high expression levels of these prognostic FRGs could increase the risk score of LGG.

### 2.8. Construction of an mRNA–miRNA–lncRNA Network

As shown in Figure 9A, the mRNA expression levels of the four prognostic FRGs varied for different pathologic stages and histological grades of gliomas. We also used the HPA database to identify the protein expression of HSPA5, SLC1A5, and NFE2L2. The representative IHC images and quantification results showed an upward expression trend of these FRGs regardless of tumor stage or histological grade (Figure 9B–G). These results suggest that the four prognostic FRGs could be involved in tumor progression in LGG. 

To clarify the potential molecular mechanism, we subsequently constructed a network of mRNA–miRNA–lncRNA interactions. According to TarBase V.8, we identified hsa-miR-124-3p, hsa-miR-155-5p, and hsa-miR-16-5p as the targeting miRNAs, which could bind to the four prognostic FRGs (Figure 10A). Further analysis revealed that only miR-124-3p was negatively correlated with the four prognostic FRGs in LGG (Figure 10B). Therefore, we explored the upstream lncRNA targeting miR-124-3p to construct the miRNA–lncRNA axis. As shown in Figure 10C, five lncRNAs (LINC00240, NEAT1, KCNQ1OT1, XIST, and MALAT1) that bind to miR-124-3p were identified. Correlation analysis revealed that LINC00240, NEAT1, and MALAT1 were negatively correlated with miR-124-3p in LGG (Figure 10D). However, only NEAT1 could reduce the LGG patients’ survival probability (Figure 10E–G). Thus, the NEAT1/miR-124-3p/four prognostic FRGs (LPCAT3, SLC1A5, HSPA5, and NFE2L2) regulatory axis was constructed and may play a vital role in the progression of LGG (Figure 10H).

## 3. Discussion

The aberrance of regulated cell death (RCD) can significantly promote tumorigenesis and tumor development. For example, resistance to apoptosis is a hallmark of cancer [14]. Therefore, the mechanisms of nonapoptotic cell death have been identified and clarified over decades to improve cancer diagnosis and treatment. As a novel iron-dependent form of RCD, ferroptosis is characteristically distinct from apoptosis, autophagy, and pyroptosis [15]. Recently, ferroptosis has been reported to play dual roles in the tumorigenesis of human malignancies. On the one hand, ferroptosis can induce tumor cell death, and could provide a potential prognostic and therapeutic target for cancer [16,17]. On the other hand, ferroptosis can trigger inflammation-associated immunosuppression and promote tumor growth. However, the potential role of ferroptosis in the LGG immune microenvironment remains elusive. In this study, we systemically investigated the clinical relations, TME features, and potential regulatory axis of ferroptosis patterns in LGG.

Using the expression profiles of FRGs, we applied consensus clustering analysis and classified LGG patients into two clusters. Previous studies showed that the integrating of multidimensional biological data and clinical features can subdivide highly heterogeneous cancers into more accurate subtypes for individualized treatment [18]. For example, based on the autophagy features, colorectal cancer patients were classified into high- and low-risk groups. More aggressive and targeted therapies were required in the high-risk group [19]. Based on the expression signatures of immune-related genes, lung adenocarcinoma patients were divided into two prognostically and clinically relevant subtypes. Consequently, the patients from the high-risk subtype were more responsive to immune checkpoint blockade treatment [20]. In our study, poor prognosis and upregulated expression of CD274 were observed in cluster 2 patients, and further analysis revealed that cluster 2 harbored significantly higher infiltrated levels of CD8+T cells. Normally, antigen-specific T cells play pivotal roles in eliminating tumor cells. However, tumor cells could upregulate the expression of CD274 to exert negative regulatory effects and blunt the anti-tumor function of T cells [21,22]. In addition, higher infiltration levels of macrophages were also observed in cluster 2. As the innate immune sentinels and host immune barriers, macrophages are the predominant component in glioma [23] and can engulf cancer cells. However, it was reported that when macrophages are recruited to the glioma environment, they can be reprogramed and repolarized to release a wide array of cytokines and growth factors, which facilitate tumor proliferation, survival and migration [24,25]. Therefore, even though there are more immune cells infiltrating into the tumor, most of them are in a state of immunosuppression or promoting tumor development, which results in the poor prognosis of cluster 2 patients.

Given the vital roles of ferroptosis in LGG TME, we selected 25 FRGs and identified four adverse prognostic regulators (LPCAT3, SLC1A5, HSPA5, and NFE2L2) as the key FRGs, which were elevated in tumors and were positively correlated with CD274 expression. We then detected their mRNA expression levels in glioma cells and normal astrocyte cells. The results of qRT-PCR confirmed that these genes were upregulated in glioma cells and could influence the CD274 mRNA expression. Based on these four prognostic FRGs, we constructed a novel prognostic risk model with high sensitivity and specificity, which could contribute to the early diagnosis of LGG. According to the risk score, LGG patients were stratified into high-risk and low-risk subgroups, and the high-risk subgroup patients showed a shorter OS and worse prognosis. 

In addition, we investigated potential therapeutic targets for these patients. It has been reported that ncRNAs (miRNAs, lncRNAs, and circRNAs) could participate in the regulation of gene expression through the competing endogenous RNA mechanism [26,27]. Based on prediction programs and correlation analysis, miR-124-3p was selected as the upstream tumor suppressive miRNA with the most potential among the four prognostic FRGs. Previous studies showed that miR-124-3p played inhibitory roles in modulating proliferation and invasion of gliomas [28,29]. This was consistent with our findings. Next, we predicted three upstream lncRNAs of miR-124-3p. Based on the ceRNA hypothesis, the potential lncRNAs should be oncogenic in LGG. By conducting survival analysis and correlation analysis, NEAT1 was identified as the upregulated lncRNA with the most potential. NEAT1 was reported to function as an oncogene and could promote glioma progression [30,31]. Above all, we developed robust FRGs signatures for diagnosis as well as predicting the outcome of LGG patients. In addition, the NEAT1/miR-124-3p/4 prognostic FRGs (LPCAT3, SLC1A5, HSPA5, and NFE2L2) axis was identified as potential regulatory pathways in LGG.

However, the potential limitations of this study should be mentioned. First, according to a previous study, we selected 25 key ferroptosis regulators for analysis. It is probable that other potential ferroptosis regulators were left out of our analysis. Secondly, due to insufficient data in our own cohort, we only analyzed the clustering subtypes as well as the interactions between the TME and ferroptosis regulators based on the TCGA database. As such, larger cohorts with further reliable validation need to be analyzed in the future. Thirdly, we only validated the impact of four prognostic FRGs on CD274 at the transcriptomic level, rather than the protein level. Therefore, the functions, specific regulatory mechanisms on CD274, and potential drug targets of the four prognostic FRGs need to be further revealed.

In summary, we systematically analyzed the expression profiles and prognosis indicators of FRGs, the correlations with CD274, and their role in LGG TME. By consensus clustering, two independent clusters with obvious tumor heterogeneity, distinct CD274 expression, and a diverse TME landscape were established. This could contribute to the risk stratification and precision therapy for patients with LGG. Among the selected FRGs, we identified four key genes (LPCAT3, SLC1A5, HSPA5, and NFE2L2) as prognostic FRGs for LGG patients. Further in vitro experiments confirmed their high expression levels and correlations with CD274. We subsequently developed a novel prognostic model for the diagnosis of LGG with high sensitivity. Further ceRNA analysis demonstrated that the NEAT1/miR-124-3p/4 prognostic FRGs (LPCAT3, SLC1A5, HSPA5, and NFE2L2) regulatory axis could be a novel and potential therapeutic target. Certainly, a series of clinical trials and further basic research are needed to help determine the targets for enhancing the efficacy of cancer immunotherapy.

## 4. Materials and Methods

### 4.1. Data Acquisition

RNA-sequencing expression (level 3) profiles and corresponding clinical information for LGG were downloaded from the Cancer Genome Atlas (TCGA; https://portal.gdc.com (accessed on 1 February 2023)) database. The data included 513 LGG samples. Normal tissue samples were obtained from Genotype-Tissue Expression Project (GTEx; https://www.gtexportal.org/home/datasets (accessed on 1 February 2023)) database. The Gene Expression Profiling Interactive Analysis (GEPIA; http://gepia.cancer-pku.cn/ (accessed on 9 February 2023)) database was used to determine the differential expression of FRGs in LGG and normal tissue samples. The GSE16011 dataset was downloaded from the Gene Expression Omnibus (GEO; http://www.ncbi.nih.gov/geo (accessed on 9 February 2023)) database and was used to further validate the expression levels of the four prognostic FRGs. The Human Protein Atlas (HPA; http://www.proteinatlas.org/ (accessed on 1 February 2023)) database was used to determine the differential expression of SLC1A5, HSPA5, and NFE2L2 at the protein level. The Tumor Immune Single-cell Hub 2 (TISCH2; http://tisch.comp-genomics.org/ (accessed on 1 February 2023)) database was used to decipher the TME heterogeneity between IDH-wildtype and IDH-mutant glioma tumors and determine the expression levels of the four prognostic FRGs at the single-cell level.

### 4.2. Subgroup Analysis

The R software ‘Consensus Cluster Plus’ (v1.54.0) package was adopted for consistency analysis of the expression levels of FRGs in LGG patients. The maximum number of clusters was 6, and 80% of the total sample was drawn 100 times (clusterAlg = ‘hc’, innerLinkage = ‘ward.D2’). The R software package pheatmap (v1.0.12) was used to draw clustering heatmaps. The gene expression heatmap retained genes with SD > 0.1. If the number of input genes was more than 1000, it extracted the top 25% genes after sorting the SD. 

### 4.3. Co-Expression Analysis

The correlation pheatmap between the FRGs and immune checkpoint genes were displayed by the R software package. Spearman’s correlation analysis was used to describe the correlation between quantitative variables without a normal distribution. *p* < 0.05 was considered statistically significant.

### 4.4. Immune Score Analysis

To assess the reliability of the results of the immune score evaluation, we used ‘CIBERSORT’, which is the latest algorithm of the ‘immunedeconv’ package. All the analysis methods and R packages used in immune score analysis were implemented by R software packages ‘ggplot2’ and ‘pheatmap’.

### 4.5. Survival Analysis

For the Kaplan–Meier curves, *p*-values and hazard ratios (HR) with 95% confidence interval (CI) were generated by log-rank tests and univariate cox proportional hazards regression. A log-rank test was used to compare differences in survival between different groups. A timeROC analysis was used to compare the predictive accuracy of the 4 prognostic FRGs and risk scores. The LASSO regression algorithm was used for feature selection with 10-fold cross-validation. The R package ‘glmnet’ was used for the analysis. *p* < 0.05 was considered statistically significant.

### 4.6. Competing Endogenous RNA Network Construction

To clarify the potential function of the 4 prognostic FRGs in LGG, a competing endogenous RNA (ceRNA) network was constructed. TarBase V.8 (https://dianalab.e-ce.uth.gr/html/diana/web/index.php?r=tarbasev8 (accessed on 1 February 2023)) was utilized to predict the miRNA targets binding to the 4 prognostic FRGs. Based on the identified miRNAs, StarBase v2.0 (https://starbase.sysu.edu.cn/starbase2/index.php (accessed on 1 February 2023)) and ENCORI (https://starbase.sysu.edu.cn/ (accessed on 1 February 2023)) were utilized to predict lncRNA targets which could interact with miRNAs. Using the TCGA database, the expression and prognostic value of these miRNAs and lncRNA targets were also explored in LGG patients. All analyses were considered statistically significant at *p* < 0.05.

### 4.7. Cell Culture

Human glioma cells (T98G and DBTRG-05MG) and normal astrocytes cells (HA1800) were purchased from the American Type Culture Collection (ATCC, Manassas, VA, USA). All the cells were cultured in DMEM media (Yeasen, Shanghai, China) supplemented with 10% fetal bovine serum (Yeasen, Shanghai, China) at 37 °C in an atmosphere of 5% CO_2_.

### 4.8. Plasmids and Transfection

According to the mRNA sequences in GenBank, we designed 2 different small haircut RNA (shRNA) sequences by using the online design software of BLOCK-iT™ RNAi Designer (http://rnaidesigner.thermofisher.com/rnaiexpress/ (accessed on 9 February 2023)) for each mRNA sequence. The nucleotide sequences are shown in Appendix A. Human glioma cells were plated at a density of 4 × 10^5^ per 60 mm dish for 18 h before transfection. The transfection was performed using ExFect transfection reagent (Vazyme, Nanjing, China) according to the manufacturer’s instructions.

### 4.9. Quantitative Real-Time PCR

Total RNA was extracted using Trizol reagent (Invitrogen, Waltham, MA, USA) according to the manufacturer’s instructions. We used 1st strand cDNA Synthesis kit (Yeason, Shanghai, China) to synthesize the cDNA and assessed the RNA concentration using a microplate reader (Biotek Take 3, Winooski, VT, USA). The qRT-PCR for cDNA was performed on Hieff^®^ qPCR SYBR^®^ Green Master Mix (Yeason, Shanghai, China). Beta-actin was used as an endogenous normalization reference. All the primers are shown in Appendix A.

### 4.10. Quantitation of IHC Images

All the IHC images were assessed using the IHC Profiler plugin [32], based on ImageJ software (v1.8.0). The staining intensity (average gray value) and staining area (positive area percentage) of positive cells were measured and were used as indices. Finally, the IHC images were given two grades of scores as positive and negative.

## 5. Conclusions

Our study combined FRGs with CD274 to explore the immunotherapy of LGG. We performed consensus clustering analysis to determine the independent subtypes. Four dominant genes were identified as the potential prognostic FRGs. Through Lasso analysis, a risk model was established with excellent potential to predict prognosis. Furthermore, a competing endogenous RNA network was constructed to provide a potential therapeutic target for LGG.

## Figures and Tables

**Figure 1 ijms-24-12880-f001:**
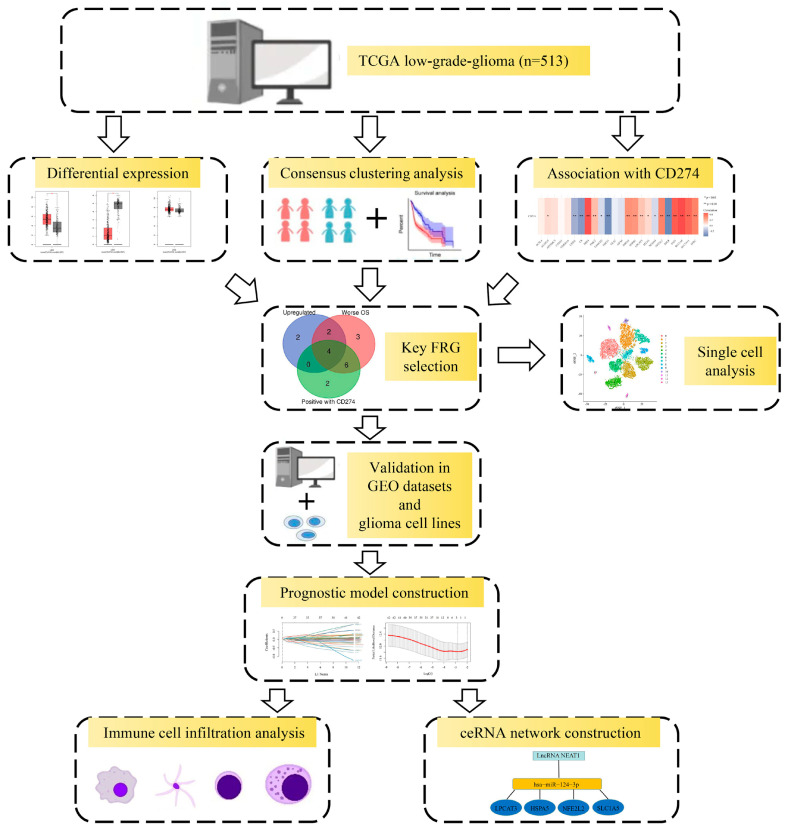
The workflow of the entire study.

**Figure 2 ijms-24-12880-f002:**
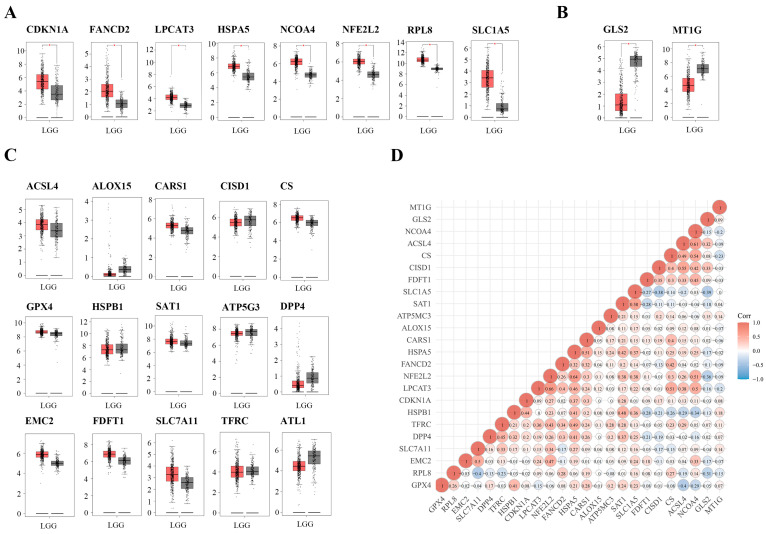
The expression distribution and correlation of FRGs in low-grade glioma (LGG) patients. (**A**–**C**) The expression levels of 25 FRGs in LGG were compared with corresponding GTEx normal tissues (tumor in red, *n* = 518, and normal in grey, *n* = 207). (**D**) Correlations between the expression of FRGs. * *p* < 0.05.

**Figure 3 ijms-24-12880-f003:**
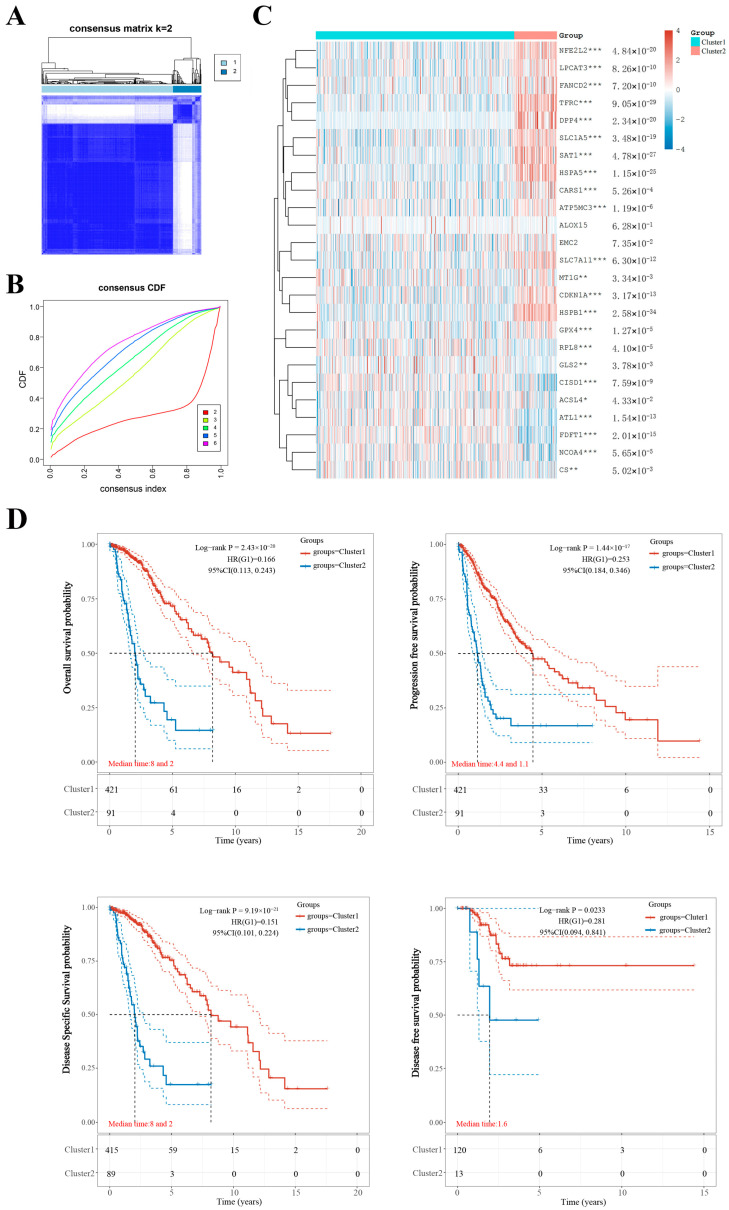
Differential expression pattern of FRGs and survival in two low-grade glioma (LGG) clusters. (**A**) Consensus clustering matrix for k = 2. (**B**) Cumulative distribution function curves for k = 2 to 6. (**C**) Heat map visualizing the expression patterns of FRGs in two LGG clusters. (**D**) The Kaplan–Meier curves show the overall survival, progression free survival, disease specific survival, and disease-free survival for two clusters of LGG patients. * *p* < 0.05, ** *p* < 0.01, and *** *p* < 0.001. Expression distribution and correlation of FRGs in low-grade glioma (LGG) patients.

**Figure 4 ijms-24-12880-f004:**
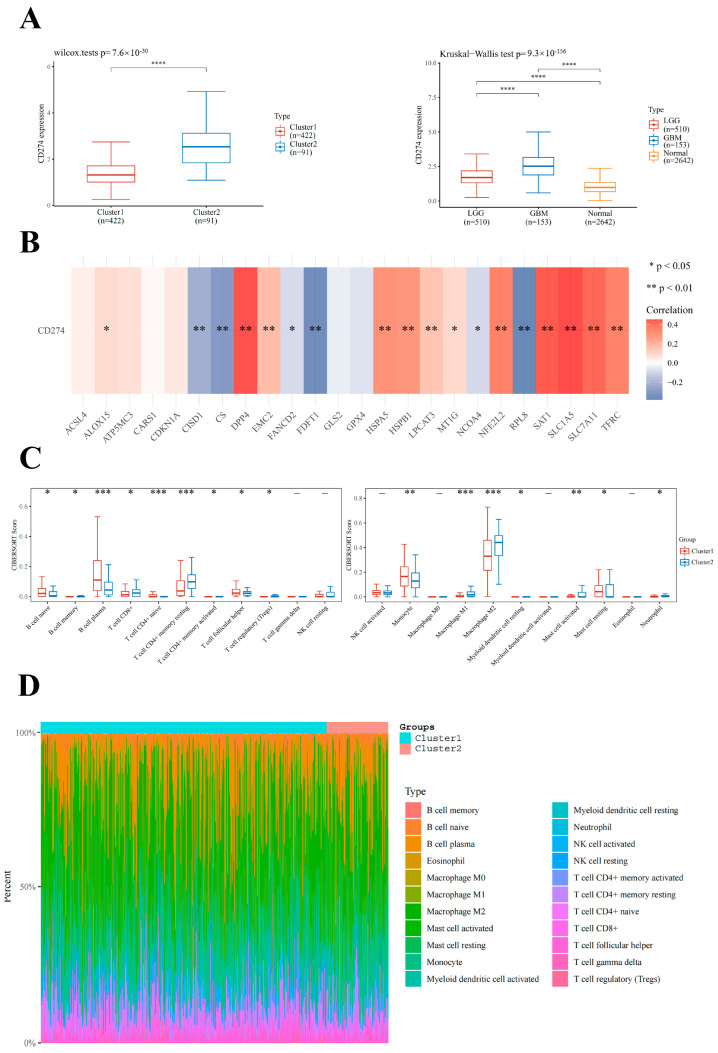
Correlation of FRGs with CD274 and tumor immune cells infiltration in LGG. (**A**,**B**) The expression levels of CD274 in LGG/normal tissues and cluster 1/2 in LGG. (**C**) The correlation of CD274 with FRGs in the TCGA-LGG cohort. (**D**) The infiltration levels of various immune cell types in two clusters in the TCGA-LGG cohort. * *p* < 0.05, ** *p* < 0.01, *** *p* < 0.001, and **** *p* < 0.0001.

**Figure 5 ijms-24-12880-f005:**
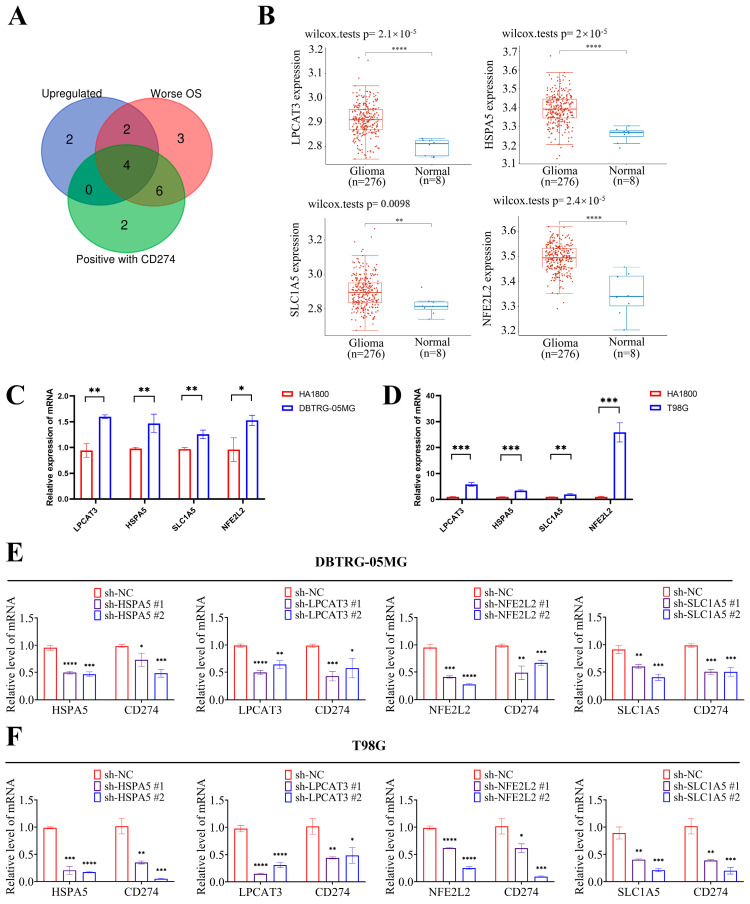
Analysis of key FRG expression in glioma cells. (**A**) The Venn diagram suggests that the up-regulation of LPCAT3, SLC1A5, HSPA5, and NFE2L2 are unfavorable prognostic factors and are positively correlated with CD274 expression in LGG. (**B**) Validation sets showed that LPCAT3, SLC1A5, HSPA5, and NFE2L2 were highly expressed in glioma tissues compared with adjacent normal tissues in GSE16011. (**C**,**D**) Relative expressions of LPCAT3, SLC1A5, HSPA5, and NFE2L2 were detected by qRT-PCR in DBTRG-05MG (**C**) and T98G (**D**) compared with normal astrocyte cells. (**E**,**F**) Glioma cells DBTRG-05MG (**E**) and T98G (**F**) were transfected with relative shRNAs or negative control shRNA. The depletion of the four key FRGs reduced CD274 mRNA expression levels. * *p* < 0.05, ** *p* < 0.01, *** *p* < 0.001, and **** *p* < 0.0001.

**Figure 6 ijms-24-12880-f006:**
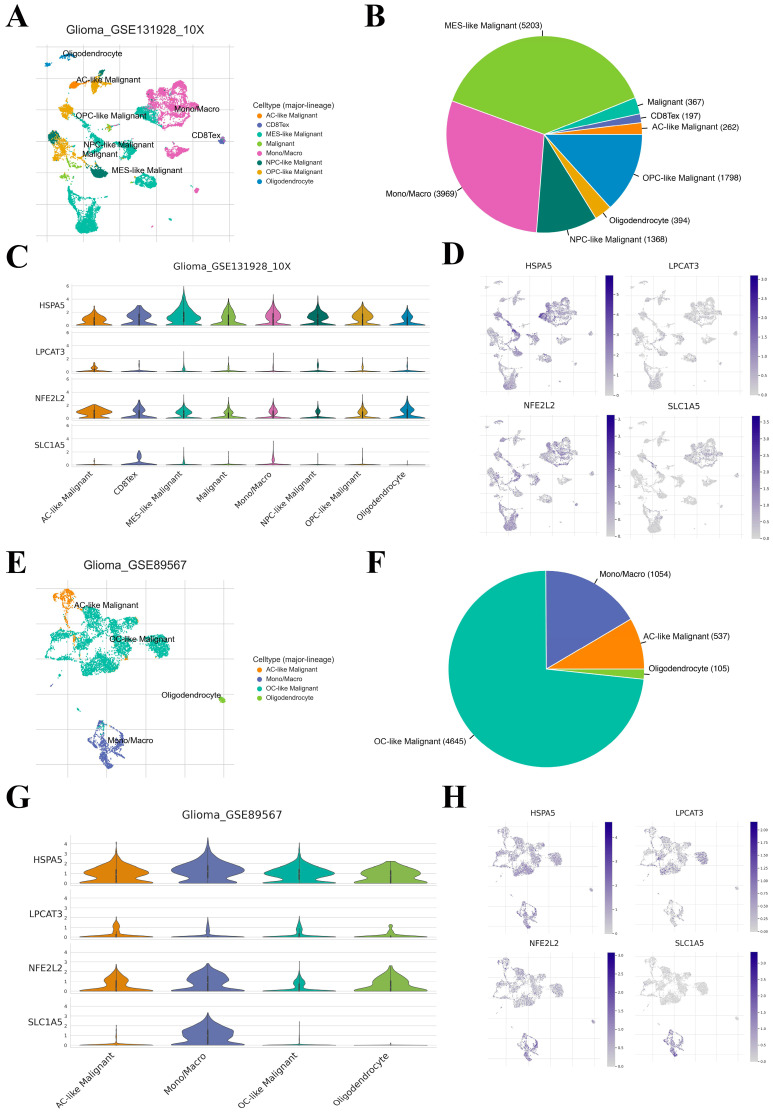
Correlation between FRGs and the TME. (**A**,**B**) The cell types and their distribution in the Glioma_GSE131928_10X dataset, based on the TISCH2 database. (**C**,**D**) The distribution and expression levels of four key FRGs (LPCAT3, SLC1A5, HSPA5, and NFE2L2) in different cell types were analyzed using single-cell resolution in the Glioma_GSE131928_10X dataset. (**E**,**F**) The cell types and their distribution in the Glioma_GSE89567 dataset. (**G**,**H**) The distribution and expression levels of four key FRGs in the Glioma_GSE89567 dataset.

**Figure 7 ijms-24-12880-f007:**
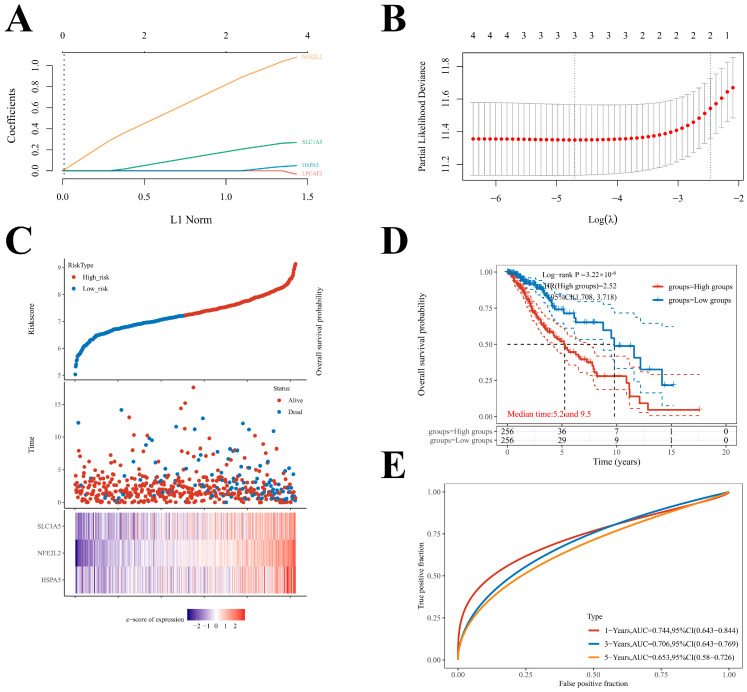
Construction of a prognostic FRG model. (**A**) LASSO coefficient profiles of four key FRGs. (**B**) Plots of the 10-fold cross-validation error rates. (**C**) Distribution of risk score, survival status, and the expression of four prognostic FRGs in LGG. (**D**,**E**) Overall survival curves for LGG patients in the high-risk and low-risk groups and the ROC curve of measuring the predictive value.

**Figure 8 ijms-24-12880-f008:**
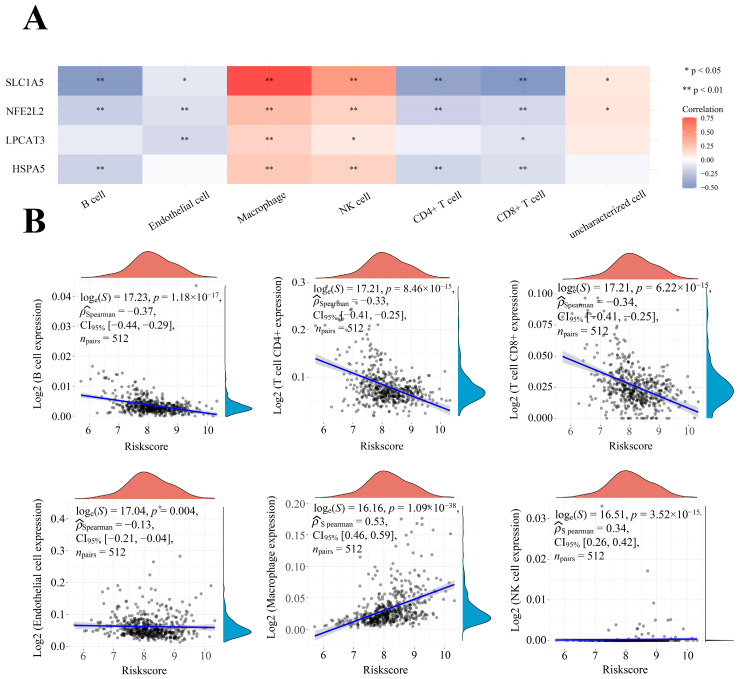
The association between four prognostic FRGs and immune infiltration. (**A**) The association between the abundance of immune cells and the expression levels of LPCAT3, SLC1A5, HSPA5, and NFE2L2. (**B**) The correlation between four prognostic FRGs and risk score in LGG.

**Figure 9 ijms-24-12880-f009:**
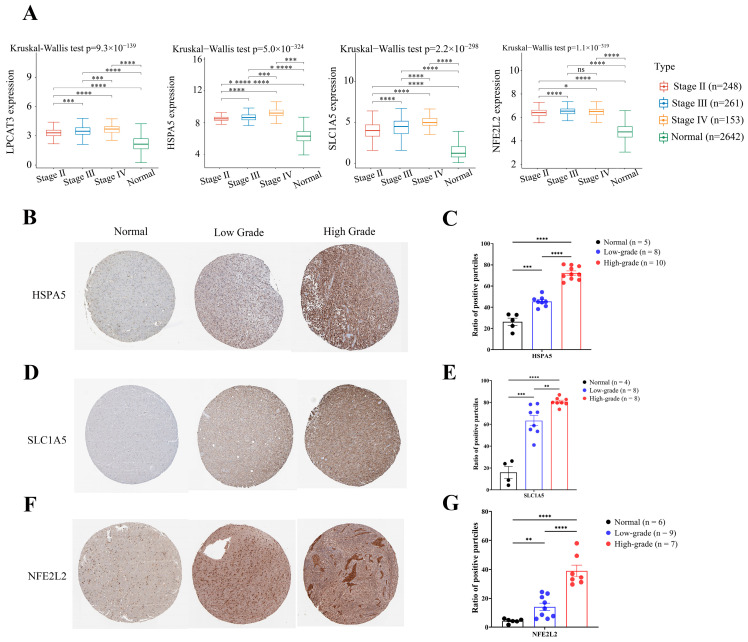
Analysis of four prognostic FRGs’ expression in different tumor stages. (**A**) mRNA expression levels of LPCAT3, HSPA5, SLC1A5, and NFE2L2. (**B**,**D**,**F**) Representative immunohistochemical staining showing the protein expression of HSPA5, SLC1A5, and NFE2L2 in cerebral cortex, low-grade glioma, and high-grade glioma from the HPA database. (**C**,**E**,**G**) The quantification results of HSPA5, SLC1A5, and NFE2L2 protein expression in IHC images. * *p* < 0.05, ** *p* < 0.01, *** *p* < 0.001, and **** *p* < 0.0001.

**Figure 10 ijms-24-12880-f010:**
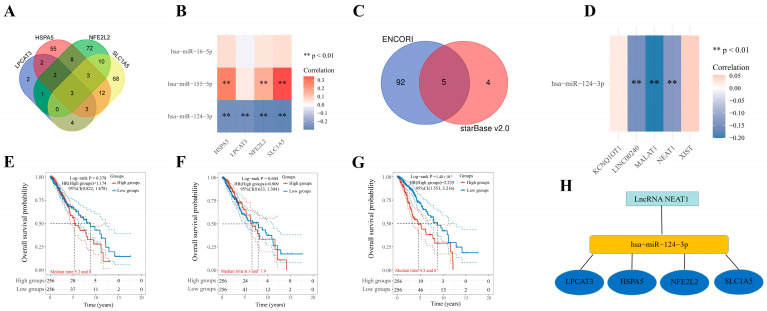
Construction of ceRNA network. (**A**) Results of miRNA target predicted by TarBase V.8. (**B**) The correlation between four prognostic FRGs and predicted miRNAs. (**C**) Results of lncRNA targets predicted by ENCORI and StarBase V2.0. (**D**) The correlation between predicted miRNAs and lncRNAs. (**E**–**G**) The prognostic value of LINC00240 (**E**), MALAT1 (**F**), and NEAT1 (**G**) in LGG. (**H**) The network of lncRNA–miRNA–mRNA.

**Table 1 ijms-24-12880-t001:** Clinical characteristics of two clusters of LGG patients.

	Feature	Cluster 1	Cluster 2	*p*-Value
Status	Alive	344	44	<0.0001
	Dead	78	47	
Age	Mean (SD)	41.3 (12.7)	50.7 (13.7)	<0.0001
	Median [Min Max]	39 [14 74]	53 [24 87]	
Gender	Female	189	39	0.826
	Male	233	52	
Race	Asian	6	2	0.145
	Black	14	7	
	White	392	81	
	American Indian		1	
Grade	Discrepancy	1		<0.0001
	G2	234	15	
	G3	187	76	
Radiation therapy	Non-radiation	113	8	<0.0001
	Radiation	101	41	

## Data Availability

The data applied to support the results of this study are available in the TCGA database https://portal.gdc.com (accessed on 1 February 2023), GTEx database https://www.gtexportal.org/home/datasets (accessed on 1 February 2023), GEPIA database http://gepia.cancer-pku.cn/ (accessed on 9 February 2023), GEO database http://www.ncbi.nih.gov/geo (accessed on 9 February 2023), HPA database http://www.proteinatlas.org/ (accessed on 1 February 2023), TISCH2 database http://tisch.comp-genomics.org/ (accessed on 1 February 2023), TarBase V.8 database https://dianalab.e-ce.uth.gr/html/diana/web/index.php?r=tarbasev8 (accessed on 1 February 2023), StarBase v2.0 database https://starbase.sysu.edu.cn/starbase2/index.php (accessed on 1 February 2023), and ENCORI database https://starbase.sysu.edu.cn/ (accessed on 1 February 2023).

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
