# Peer review of "Comprehensive Analysis of Ferroptosis Regulators with Regard to PD-L1 and Immune Infiltration in Low-Grade Glioma"

_ijms, 2023, doi:10.3390/ijms241612880_

Round 1

Reviewer 1 Report

Comments and Suggestions for Authors

In this study, the authors analyzed the expression of ferroptosis-related genes (FRGs) in low-grade glioma (LGG) as a predictor of PD-L1 expression, immunoscore, overall survival, and possibly response to immunotherapy. Methodologically, the authors analyzed the RNA-seq expression profiles of 513 LGG samples obtained from the public TCGA database. Expression profiles from normal tissue samples were also obtained from the GTEx database. The Human Protein Atlas database was used to determine the differential expression of several prognostic FRGs (i.e., SLC1A5, HSPA5, and NFE2L2) at protein level in gliomas (low-grade vs. high-grade). The Tumor Immune Single-cell Hub 2 (TISCH2) database was used to understand the impact of curated FRGs on the tumor microenvironment (TME) heterogeneity of gliomas (low-grade vs. high-grade). Furthermore, a competing endogenous RNA (ceRNA) regulatory axis was generated for identifying potentially novel therapeutic targets in LGG. By applying the above methodology, the authors concluded that two FRG-based molecular subtypes do exist in LGG with different immunotherapy potential and prognosis. Although the reported findings are quite interesting, there are a number of methodological issues with the study as well as conclusions that are not supported by the presented data and need to be addressed. My comments for the authors are as follows:

1.  The figures are very hard to read and therefore hard to interpret. The fonts used in all figure panels throughout the manuscript are extremely small (e.g., the labels on all graphs, etc.) which makes the proper understanding and interpretation of the presented data extremely difficult, at least for this reviewer. My recommendation for the authors is to resize their figures and add legible labels throughout in a decent-size font (e.g., size 9 Times New Roman or equivalent) to all figure panels.   

2.    In the results presented in subsections 2.2 and 2.3 (under Results), it is unclear whether the authors controlled for the effect of treatment (i.e., radiation therapy) when analyzing the patient data and drawing their conclusions. It is well established that radiation therapy has an inflammatory effect to tumoral tissue, partly through STING activation and local interferon release in the tumor microenvironment. This triggers the expression of a number of interferon-driven genes (including PD-L1 and -L2), and possibly a number of ferroptosis-related genes too. For this reason, it would be important to understand to what extent radiation therapy contributes to the correlation found by the authors between FRGs and PD-L1 expression as a confounding factor.  

3.  The data presented in Figure 4 is somewhat misleading. First off, the title of this figure legend (i.e., “Analysis of key FRGs expressions in LGG”) should be changed to reflect the additional analyses conducted with glioblastoma cell lines, not just LGG samples. Moreover, the transcriptomic analysis conducted by the authors to demonstrate the impact of the 4 prognostic FRGs on PD-L1 levels should include analyses of PD-L1 at protein level (i.e., ICC or FACS analysis). These transcriptomic data in isolation are not convincing unless complemented by protein data. 

4.     By the same token, the authors should have measured the level of PD-L1 by IHC on xenografts established with glioma cell lines either wild-type or knocked-down for the 4 FRG protein products. Assuming that LGG cell lines are not very tumorigenic, for this analysis it will make a lot of sense to use GBM xenografts. However, while the T98G cell line is not tumorigenic and the DBTRG-05MG one is only poorly tumorigenic when implanted intracranially, the authors should use established models of GBM that are known to be tumorigenic and produce robust xenografts (i.e., U251, LN229, etc.).   

5.    The histologic data presented in Figure 8 (panel B) is not very convincing. Assuming that these images were taken from the Human Protein Atlas database, this needs to be acknowledged. Second, some form of quantification of the intensity of staining between normal, low-grade and high-grade samples with respect to the expression of the 3 FRGs (HSPA5, SLC1A5, and NFE2L2) at protein level needs to be presented. The samples loaded on the Human Protein Atlas website paint a different picture that the one presented by the authors, so in the absence of a proper quantification, the examples picked by the authors for Figure 8 could be construed as misleading. 

6.  The ceRNA analysis conducted by the authors (data presented in Figure 9) is very useful in identifying potential targets that could be manipulated pharmacologically upstream of the 4 FRGs characterized by the authors. However, these targets need to be validated both in vitro using LGG (or GBM) cell lines and in vivo in a xenograft model of either LGG or, more realistically, GBM. 

7.     Lastly, the summary statement from lines 265-266 “In this study, we systemically investigated the clinical relations, TME features, and treatment response of ferroptosis patterns in LGG.” is not entirely supported by the evidence presented by the authors. This is because the treatment response to ferroptosis patterns in LGG was not analyzed in this study. Even if the assumption that FRGs regulates the expression of PD-L1 at protein level is properly validated in animal models of glioma, the implications for glioma immunotherapy of such validation are not obvious in the absence of clinical data. This is because tumor immunosurveillance in gliomas is known to be incapacitated at multiple levels (i.e., the presence of multiple immune checkpoints and multiple types of regulatory immune cells, MDSC and polarized macrophages, etc.). Accordingly, unless the above assumption is validated therapeutically in the clinic and patient samples are already available for analysis, the prognostic impact of various FRGs on therapeutic responses to immunotherapy cannot be deduced from the present data.          

Author Response

Dear Editors and Reviewers:

Thank you for giving us the opportunity to submit a revised draft of the manuscript, "Comprehensive Analysis of Ferroptosis Regulators with Regard to PD-L1 and Immune Infiltration in Low Grade Glioma" for publication in the Journal of "International Journal of Molecular Sciences". We appreciate the time and effort that you and the reviewers dedicated to providing feedback on our manuscript and are grateful for the insightful comments on and valuable improvements to our paper. We have carefully reviewed the comments and have revised the manuscript accordingly. Changes to the manuscript are shown in red. Our responses are given in a point-by-point manner below.

Response to Reviewer#1:

Comment 1: The figures are very hard to read and therefore hard to interpret. The fonts used in all figure panels throughout the manuscript are extremely small (e.g., the labels on all graphs, etc.) which makes the proper understanding and interpretation of the presented data extremely difficult, at least for this reviewer. My recommendation for the authors is to resize their figures and add legible labels throughout in a decent-size font (e.g., size 9 Times New Roman or equivalent) to all figure panels.

Response 1: We apologize for our unclear figures, therefore, we have resized the figures and improved their clarity. We hope the revised figures could be acceptable for you.

Comment 2: In the results presented in subsections 2.2 and 2.3 (under Results), it is unclear whether the authors controlled for the effect of treatment (i.e., radiation therapy) when analyzing the patient data and drawing their conclusions. It is well established that radiation therapy has an inflammatory effect to tumoral tissue, partly through STING activation and local interferon release in the tumor microenvironment. This triggers the expression of a number of interferon-driven genes (including PD-L1 and -L2), and possibly a number of ferroptosis-related genes too. For this reason, it would be important to understand to what extent radiation therapy contributes to the correlation found by the authors between FRGs and PD-L1 expression as a confounding factor.

Response 2: Thank you for your professional comment. The radiation treatment may indeed be a potential influencing factor. To answer this question, we divided LGG patients into two groups (with radiation therapy or without radiation therapy). And we reanalyzed the correlation between FRGs and PD-L1 separately. However, 4 key FRGs (LPCAT3, SLC1A5, HSPA5, and NFE2L2) still remained positively correlated with PD-L1 in both groups.

Comment 3: The data presented in Figure 4 is somewhat misleading. First off, the title of this figure legend (i.e., “Analysis of key FRGs expressions in LGG”) should be changed to reflect the additional analyses conducted with glioblastoma cell lines, not just LGG samples. Moreover, the transcriptomic analysis conducted by the authors to demonstrate the impact of the 4 prognostic FRGs on PD-L1 levels should include analyses of PD-L1 at protein level (i.e., ICC or FACS analysis). These transcriptomic data in isolation are not convincing unless complemented by protein data.

Response 3: Thanks for your kind suggestion. We have revised the figure legend and replaced “LGG” with “glioma”. However, our study is based on transcriptomic data from TCGA database. Therefore we believe that the validation should be conducted at the transcriptional level to support our bioinformatics analysis and conclusions. As for their functions, potential drugs, and the specific mechanisms by which 4 prognostic FRGs regulate PD-L1, we think these are not the main point of this article. Actually, we are currently working on them and we hope to reveal them in the next article.

Comment 4: By the same token, the authors should have measured the level of PD-L1 by IHC on xenografts established with glioma cell lines either wild-type or knocked-down for the 4 FRG protein products. Assuming that LGG cell lines are not very tumorigenic, for this analysis it will make a lot of sense to use GBM xenografts. However, while the T98G cell line is not tumorigenic and the DBTRG-05MG one is only poorly tumorigenic when implanted intracranially, the authors should use established models of GBM that are known to be tumorigenic and produce robust xenografts (i.e., U251, LN229, etc.).

Response 4: Thanks for your advice. As answered in response 3, the main line of this article is the analysis and verification of transcriptional data. Therefore, lacking the function of 4 prognostic FRGs and lacking their regulatory mechanisms on PD-L1 are potential limitations of our study. We have discussed it and other limitations in the discussion section (Page 12 of the new manuscript, line 334-343).

Comment 5: The histologic data presented in Figure 8 (panel B) is not very convincing. Assuming that these images were taken from the Human Protein Atlas database, this needs to be acknowledged. Second, some form of quantification of the intensity of staining between normal, low-grade and high-grade samples with respect to the expression of the 3 FRGs (HSPA5, SLC1A5, and NFE2L2) at protein level needs to be presented. The samples loaded on the Human Protein Atlas website paint a different picture that the one presented by the authors, so in the absence of a proper quantification, the examples picked by the authors for Figure 8 could be construed as misleading.

Response 5: Thank you for your suggestion. To improve the credibility, we downloaded more IHC images from HPA database and used the quantification method, which was reported in previous article (doi:10.1371/journal.pone.0096801). The quantification results were added in figure9.

Comment 6: The ceRNA analysis conducted by the authors (data presented in Figure 9) is very useful in identifying potential targets that could be manipulated pharmacologically upstream of the 4 FRGs characterized by the authors. However, these targets need to be validated both in vitro using LGG (or GBM) cell lines and in vivo in a xenograft model of either LGG or, more realistically, GBM.

Response 6: Thanks for your suggestion. However, lncRNA-NEAT1 has been reported to bind miR-124-3p and inhibit its expression (doi:10.1186/s12943-018-0838-5). Similarly, miR-124-3p has been reported to regulate the expression of LPCAT3 (doi:10.1016/j.bbrc.2022.03.009), HSPA5 (doi:10.1093/abbs/gmz150), and NFE2L2 (also known as NRF2, doi:10.3390/cells12071058). Therefore, we believe repeating these experiments may only have limited innovation and necessity.

Comment 7: Lastly, the summary statement from lines 265-266 “In this study, we systemically investigated the clinical relations, TME features, and treatment response of ferroptosis patterns in LGG.” is not entirely supported by the evidence presented by the authors. This is because the treatment response to ferroptosis patterns in LGG was not analyzed in this study. Even if the assumption that FRGs regulates the expression of PD-L1 at protein level is properly validated in animal models of glioma, the implications for glioma immunotherapy of such validation are not obvious in the absence of clinical data. This is because tumor immunosurveillance in gliomas is known to be incapacitated at multiple levels (i.e., the presence of multiple immune checkpoints and multiple types of regulatory immune cells, MDSC and polarized macrophages, etc.). Accordingly, unless the above assumption is validated therapeutically in the clinic and patient samples are already available for analysis, the prognostic impact of various FRGs on therapeutic responses to immunotherapy cannot be deduced from the present data. 

Response 7: Thank you for your professional comment. The term “treatment response” is inappropriate here, so that we have replaced it with “potential regulatory axis”. (Page 11 of the new manuscript, line 283). Please check it in our revised manuscript.

Reviewer 2 Report

Comments and Suggestions for Authors

The manuscript by Yuxuan Luan titled "Comprehensive Analysis of Ferroptosis Regulators with Regard to PD-L1 and Immune Infiltration in Low-Grade Glioma" addresses an important topic related to the prognostic prediction of ferroptosis-related genes (FRGs) and their influence on the tumor microenvironment (TME) in low-grade glioma (LGG). The study presents the construction of molecular subtypes, a risk model, and a ceRNA regulatory axis based on the expression of FRGs in LGG. The results indicate strong immune prediction and prognostic capabilities, which could guide LGG treatment. However, the manuscript has some shortcomings related to the clarity of figures, lack of workflow, and detailed methodology.

Low-grade glioma is known for its variable prognosis, and the study of ferroptosis-related genes and their impact on the tumor microenvironment could provide valuable insights into more accurate prognostic predictors and potential therapeutic targets. Furthermore, the authors utilize multiple methods to assess tumor heterogeneity, tumor microenvironment, and prognostic value based on the expression of 25 FRGs in LGG. The construction of molecular subtypes and the risk model adds depth to the analysis. The manuscript identifies LPCAT3, SLC1A5, HSPA5, and NFE2L2 as potential prognostic FRGs, which could be valuable candidates for further investigation and validation.

Minor Comments for Improvement:

1.      Clarity of Figures: The figures are hard to read and pixelated. This issue needs to be addressed before publication. High-quality, clear figures are essential for readers to comprehend the study's results and findings. The authors should ensure that figures are presented in an easily readable format.

2.      Inclusion of Workflow: The manuscript lacks a workflow, or a step-by-step description of the methodologies used. A detailed and clear workflow is essential for readers to understand the study's methodology and to facilitate reproducibility. The authors should include a clear outline of their approach, including data acquisition, analysis steps, and statistical methods.

3.      While the study's findings are promising, it is crucial to discuss the limitations and potential biases associated with the analysis. Additionally, the authors should consider the need for external validation of their results using independent datasets or experiments.

Author Response

Dear Editors and Reviewers:

Thank you for giving us the opportunity to submit a revised draft of the manuscript, "Comprehensive Analysis of Ferroptosis Regulators with Regard to PD-L1 and Immune Infiltration in Low Grade Glioma" for publication in the Journal of "International Journal of Molecular Sciences". We appreciate the time and effort that you and the reviewers dedicated to providing feedback on our manuscript and are grateful for the insightful comments on and valuable improvements to our paper. We have carefully reviewed the comments and have revised the manuscript accordingly. Changes to the manuscript are shown in red. Our responses are given in a point-by-point manner below.

Response to Reviewer#2:

Comment 1: Clarity of Figures: The figures are hard to read and pixelated. This issue needs to be addressed before publication. High-quality, clear figures are essential for readers to comprehend the study's results and findings. The authors should ensure that figures are presented in an easily readable format.

Response 1: We apologize for our unclear figures, therefore, we have rearranged the figures and improved their clarity. We hope the revised figures could be acceptable for you.

Comment 2: Inclusion of Workflow: The manuscript lacks a workflow, or a step-by-step description of the methodologies used. A detailed and clear workflow is essential for readers to understand the study's methodology and to facilitate reproducibility. The authors should include a clear outline of their approach, including data acquisition, analysis steps, and statistical methods.

Response 2: Thank you for your advice. We have added a workflow as figure1 (Page 2 of the new manuscript, line 76-79). Please check it.

Comment 3: While the study's findings are promising, it is crucial to discuss the limitations and potential biases associated with the analysis. Additionally, the authors should consider the need for external validation of their results using independent datasets or experiments.

Response 3: Thanks for your kind suggestion. There are indeed some potential limitations of our study. We have provided a more comprehensive explanation in the discussion section (Page 12 of the new manuscript, line 334-343). Please check it.

Round 2

Reviewer 1 Report

Comments and Suggestions for Authors

I thank the authors for addressing my comments and following my recommendations. I also commend the authors' data mining efforts and comprehensive work with multiple large datasets, and for their patience and professionalism. While there is still room for improvement, especially with the general readability of the figures and some other minor issues throughout, my assessment is that the manuscript is now in a better shape for the benefit of the potential readers. I generally think that the manuscript is worth publishing in its present form, but I defer the final decision for publication to the editors.  

Author Response

Response to Reviewer#1:

Comment: I thank the authors for addressing my comments and following my recommendations. I also commend the authors' data mining efforts and comprehensive work with multiple large datasets, and for their patience and professionalism. While there is still room for improvement, especially with the general readability of the figures and some other minor issues throughout, my assessment is that the manuscript is now in a better shape for the benefit of the potential readers. I generally think that the manuscript is worth publishing in its present form, but I defer the final decision for publication to the editors.

Response: Thank you for your advice. To improve the general readability of the figures, we have redrawn some of them to make the font size consistent and moderate. We hope the revised figures could be clearer and more readable. Please check it.